# Metformin Reduces Viability and Inhibits the Immunoinflammatory Profile of Human Glioblastoma Multiforme Cells

**Daewoo Hong [1], Regina Ambe [1], Jose Barragan [1], Kristina Marie Reyes [2] and Jorge Cervantes [2,*]**

[1] Paul L. Foster School of Medicine, Texas Tech University Health Sciences Center, El Paso, TX 79905, USA
[2] Dr. Kiran C. Patel College of Allopathic Medicine, Nova Southeastern University, Fort Lauderdale, FL 33328, USA
* Correspondence: jcervan1@nova.edu

**Abstract:** Glioblastoma (GBM) is the predominant primary malignant brain tumor. Metformin, a well-known antidiabetic medication, has emerged as a potential therapeutic candidate in the treatment of GBM. We have herein investigated two aspects of the effect of MTF on GBM cells: the effect of MTF on GBM cell viability, as previous studies have shown that MTF can selectively affect human GBM tumors; and the immunomodulatory effect of MTF on GBM, as there is evidence that inflammation is associated with GBM growth and progression. The human GBM cell line (U87) was exposed to various doses of MTF (1 mM, 20 mM, and 50 mM), followed by examination of cell viability and inflammatory mediator secretion at various time points. We observed that MTF treatment exerted a dose-response effect on glioblastoma multiforme cell viability. It also had an immunomodulatory effect on GBM cells. Our study identified several mechanisms that led to the overall inhibitory effect of MTF on human GBM. Further inquiry is necessary to gain a better understanding of how these in vitro findings would translate into successful in vivo approaches.

**Keywords:** metformin; glioblastoma multiforme; immunomodulation

## 1. Introduction

Glioblastoma (GBM) is the predominant primary malignant brain tumor, notorious for its dire prognosis, with median overall survival spanning a narrow range from 14.6 to 26.3 months in clinical investigations [1]. Its hallmark traits encompass highly invasive growth, augmented angiogenesis, and profound local immunosuppression, underpinning the challenges of effective therapeutic intervention [2,3]. A pivotal factor contributing to therapy resistance lies in the presence of brain tumor initiating cells (BTICs), a cell subset adept at self-renewal and diverse cellular differentiation [4].

Metformin, a well-known antidiabetic medication, has emerged as a potential therapeutic candidate in the treatment of GBM. Derived from the legume *Galega officinalis*, metformin (MTF) was originally intended to be utilized as an antiviral drug against influenza. Its hypoglycemic properties were discovered to be one of its side effects [5]. Since then, it has been used as the standard treatment for type 2 diabetes mellitus (T2DM) as it decreases glucose production in the liver [6]. MTF has a primary effect at the level of the cellular respiratory chain [7], which helps explain its effects on various cell types [8].

Aside from its typical use in treating T2DM and metabolic syndrome, MTF has immunomodulatory activity that reduces the production of pro-inflammatory cytokines by macrophages and neutrophils [9]. In fact, MTF's anti-inflammatory effect, regardless of diabetes status, has been shown to be beneficial in patients with COVID-19 [9–11]. MTF is also the first drug of choice for lowering glucose in diabetic patients with active tuberculosis, which is characterized by harmful inflammation that destroys granuloma architecture [12].

MTF exhibits promising anti-neoplastic effects, particularly in gliomas [13]. It has been shown to inhibit the growth of human GBM cells and enhance the therapeutic response to this neoplasia [14]. Although its exact mechanism of action is yet to be determined, numerous retrospective studies have identified a trend toward improved survival in glioblastoma patients treated with MTF [15]. A systematic review of five studies analyzing MTF's potential as an antineoplastic agent in brain tumors showed prolonged survival in primary or secondary glioblastoma patients [16]. Only one of the five studies did not demonstrate that the use of MTF was associated with overall survival or prolonged free survival, possibly due to differences in MTF dosage, duration of therapy, and patient population [17]. A very recent report showed a survival benefit with MTF use in patients with glioblastomas [18].

The primary mechanism of action of MTF involves the inhibition of complex-I within the respiratory chain, resulting in altered AMPK and mTOR signaling pathways [7,19]. Besides its antioxidant, immunomodulatory, and antiviral capabilities, MTF can induce cell cycle arrest in certain cell types [20]. An in vitro study evaluating the anti-proliferative activity of MTF on four human GBM showed statistically significant inhibition of cell viability after 24 h of MTF treatment and maximal reduction of cell viability after 72 h [21]. In all four tumor-initiating cell (TIC) cultures of GBM, MTF exerted cytostatic inhibition of growth at concentrations up to maximal inhibitory concentrations (IC50). At higher concentrations, cytotoxic effects were observed. Furthermore, MTF significantly reduced the number of cell divisions [21]. Its impact on GBM spans inhibition of proliferation, invasion, induction of apoptosis, autophagy, and differentiation of BTICs, thereby augmenting radio- or chemosensitivity while sparing mature neurons [14,19,22].

The effect of MTF on GBM cell viability, showing that MTF can selectively affect human GBM tumors, and the immunomodulatory effect of MTF on GBM cells, providing evidence that inflammation is associated with GBM growth and progression [23–25], have been reported independently. We herein investigated if these two aspects of the effect of MTF on GBM were connected. Through investigation of the effect of MTF on glioblastoma cell viability, we shall begin to explore its potential as a therapeutic agent.

## 2. Material and Methods

### 2.1. Cell Viability Assay

The human GBM cell line U87 (ATCC) at a density of $1 \times 10^5$ cells/mL was exposed to various doses of MTF (1 mM, 20 mM, and 50 mM), followed by examination of cell viability at various time points (1, 24, 48, and 72 h). The MTF doses and time points that were incorporated as part of our assessment align with previous study designs in which MTF treatment was associated with concentration-dependent growth inhibition [21].

For evaluation of cell viability, we used the LIVE/DEAD cell imaging kit (Invitrogen), which uses the incorporation of the fluorescent dyes Syto6 and propidium iodide to distinguish between live and dead cells, respectively. Images were acquired using a fluorescence microscope (Motic, Schertz, TX, USA), and the number of live vs. dead cells was processed with ImageJ (NIH, Bethesda, MD, USA). Three composite fluorescent microscopy images were taken from locations within the same well. The average of the three composite images was used to determine live vs. dead cell counts. The quantification of live or dead cells was performed blindly by two individuals to minimize human error and observer bias. Flow cytometry was also used to further assess MTF's effect on U87 cell viability using an Accuri C6 flow cytometer and was analyzed using the Accuri C6 Plus software.

### 2.2. Immune Response Evaluation

We then proceeded to perform a multiplex ELISA using the Human Cytokine 29 Plex (Millipore, Macquarie Park, Australia), which detects simultaneously 29 cytokines (EGF, G-CSF, GM-CSF, IFN-α2, IFN-γ, IL-1α, IL-1β, IL-1ra, IL-2, IL-3, IL-4, IL-5, IL-6, IL-7, IL-8, IL-10, IL-12 (p40), IL-12 (p70), IL-13, IL-15, IL-17A, IP-10, MCP-1, MIP-1α, MIP-1β, TNF-α, TNF-β, VEGF, and Eotaxin/CCL11). Supernatants were collected from U87 cell cultures following exposure to 1 mM, 20 mM, and 50 mM MTF for 1, 24, 48, and 72 h. Cell

supernatant was collected from treated cells and centrifuged to remove cellular debris, and the multiplex plate was run on a Luminex MagPix instrument (Millipore) following the manufacturer's instructions.

### 2.3. Statistical Analysis

Statistical group analysis (two-way ANOVA) was performed using GraphPad Prism 10 (GraphPad Software, San Diego, CA, USA). Statistical significance was defined as *p*-values of <0.05.

### 3. Results

#### 3.1. Metformin Treatment Has a Dose-Response Effect on Glioblastoma Multiforme Cell Viability

Our data showed that MTF decreased GBM cell viability in a dose-dependent manner. Compared to untreated cells, GBM cultures treated with 50 mM of MTF resulted in a significantly higher percentage of dead cells after 72 h (Figure 1). Cells that were treated with a higher concentration of MTF (50 mM) showed a higher percentage of dead cells compared to those that were treated with lower concentrations (i.e., 1 mM and 20 mM) at a given time point (Figure 1). The percentage of dead to live cells increased as the exposure time increased beyond 24 h in the 1 mM and 2 mM MTF-treated groups (Figure 1B). Furthermore, cell viability evaluated by flow cytometry confirmed a significant decrease in the viability of U87 cells treated with MTF for 72 h (Figure 1C). Untreated cells were used as a control.

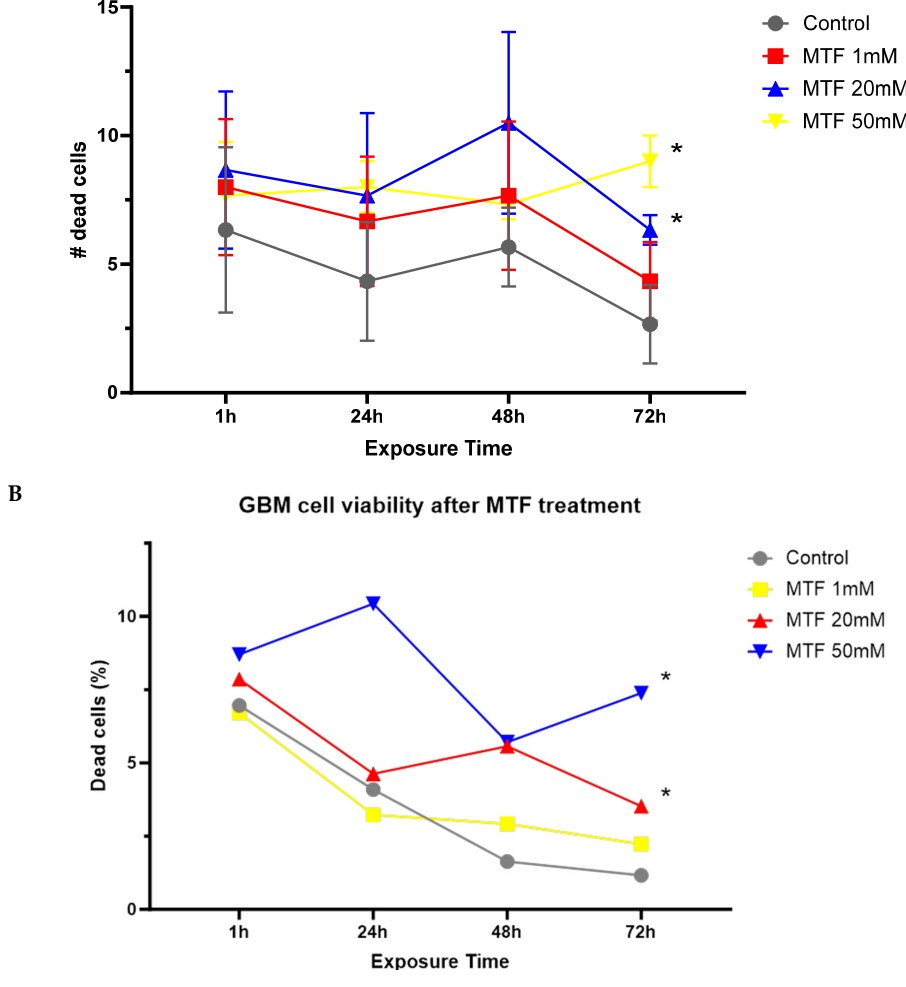

**Figure 1.** *Cont.*

C

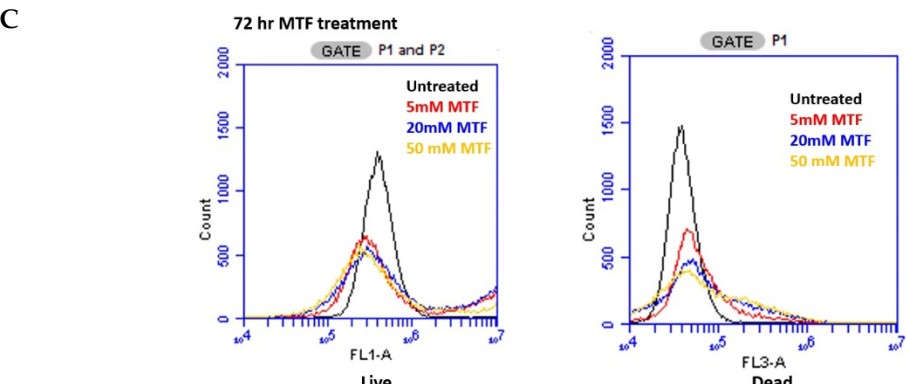

**Figure 1.** Quantitative data on the human GBM cell line (U87) (**A**) dead cell counts and (**B**) percentage of dead cells after exposure to MTF. The human GBM cell line (U87) was exposed to three different doses of MTF (1 mM, 20 mM, and 50 mM), and cell counts were performed at various time points (1, 24, 48, and 72 h). * $p < 0.001$ Two-way ANOVA. (**C**) Flow cytometry analysis of live vs. dead cells at 72 h after MTF treatment of U87 cells.

### 3.2. Metformin Has an Immunomodulatory Effect on GBM Cells

From a total of 29 analytes evaluated, we observed a reduction in the secretion of some inflammatory biomarkers upon exposure to MTF: VEGF, GM-CSF, and MCP1. VEGF in particular showed a significant reduction upon exposure to all three MTF concentrations (1 mM, 20 mM, and 50 mM) compared to the untreated control group (0 mM). This reduction in the immunoinflammatory profile of GBM cells was correlated with an increase in MTF concentration at 72 h (Figure 2A). Some others, like IL-8, IL-1b, Eotaxin, GCSF, and TNF-α, showed a reduction mainly upon exposure to 50 mM of MTF (Figure 2B).

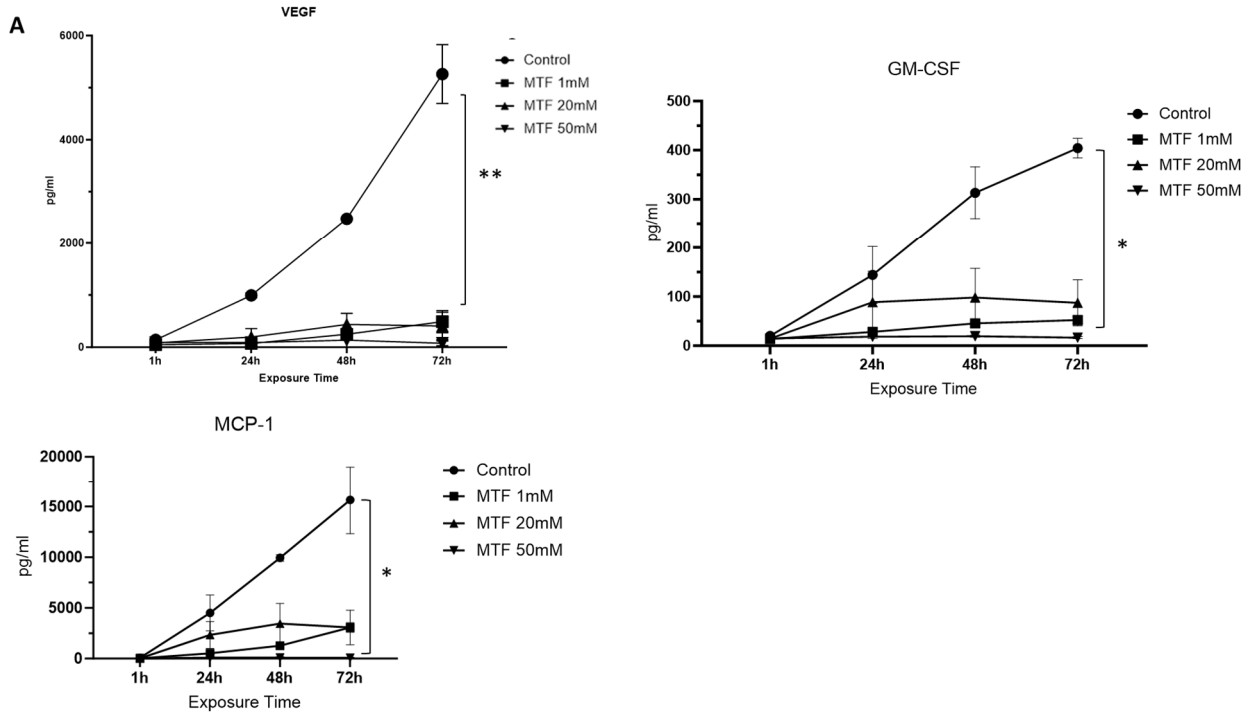

**Figure 2.** *Cont.*

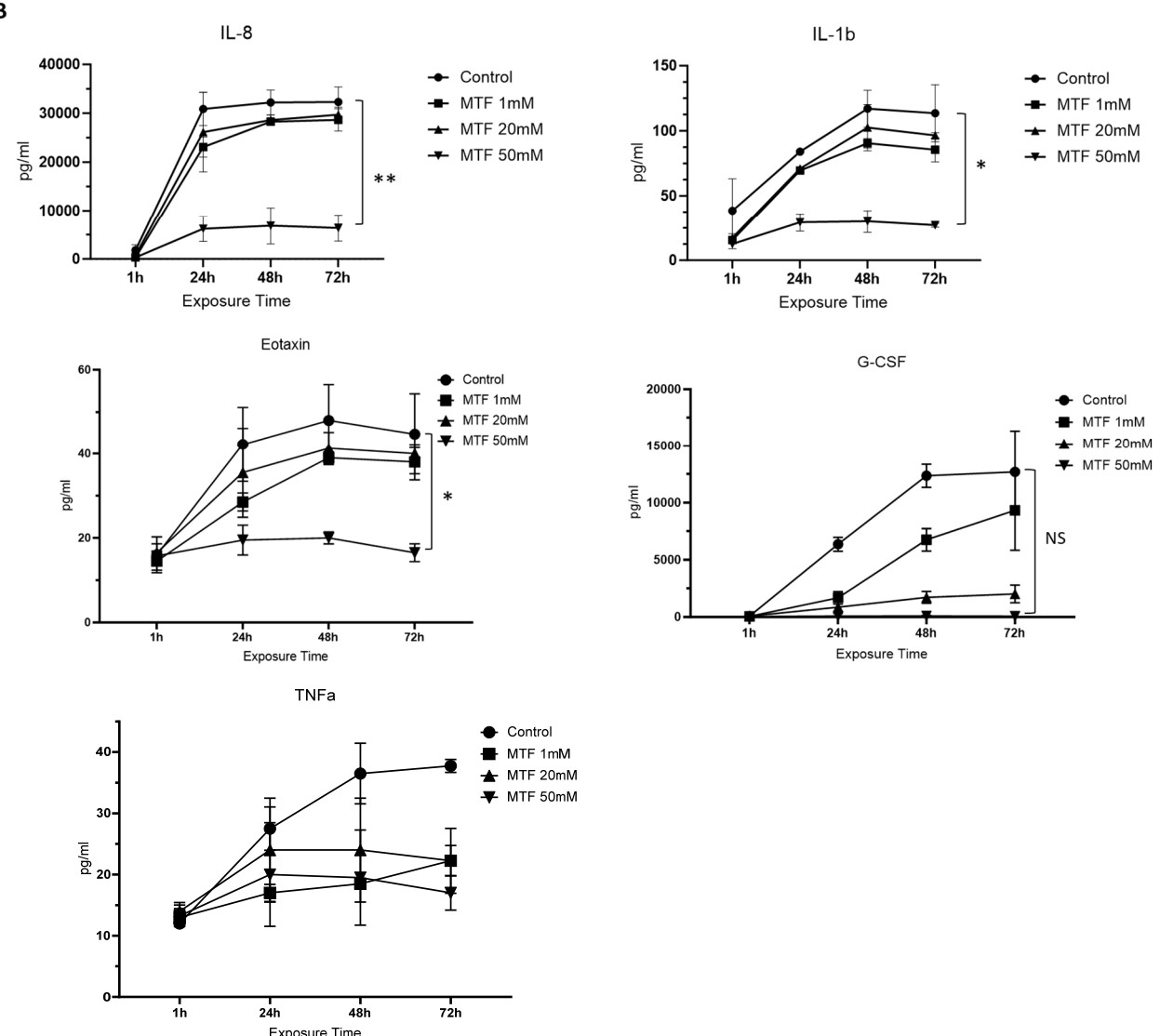

**Figure 2.** Immunomodulatory effect of MTF. Decreased secretion of mediators upon MTF exposure compared to control: (**A**) VEGF, GM-CSF, and MCP1. Decreased secretion of mediators upon exposure to 50 mM of MTF compared to control (**B**) IL-8, IL-1b, Eotaxin, G-CSF, and TNF-a. ANOVA * $p < 0.05$; ** $p < 0.001$.

## 4. Discussion

As previous studies have shown that MTF can selectively affect the growth of human GBM [14,21], we decided to explore this effect along with any immunomodulatory effect of MTF on GBM, as there is evidence that inflammation is associated with GBM growth and progression [23–25].

We observed maximal cell death at 24 h in a dose-dependent manner at that time point. Other groups have demonstrated a dose-dependent reduction of U87 cell viability and cell proliferation with MTF at doses between 1 and 50 mM for 24 h [26]. Beyond the 24-h mark, another study observed reduced cell viability in GBM TIC cultures in a time-dependent manner [21], which was the reason for us to expand our time points to 72 h. We are uncertain why cell death peaked at 48 h, leaving us to hypothesize the possibility that MTF was first able to target bulk cells and then annihilate the remaining cells by targeting GB-CSC. A point worth exploring further.

One of the major drawbacks of in vitro studies suggesting MTF's inhibitory effects on glioma cells is that the MTF doses used are significantly higher than the concentrations measured in the brains of patients [15]. Also, the dosage of MTF that was administered was much higher than a standard anti-diabetic dose [17]. In vitro studies often used MTF doses in the millimolar range [21], whereas MTF doses in the brains of diabetic patients have been measured in the micromolar range [27]. Optimal dosing regimens of MTF as an adjunct antineoplastic agent are still under investigation. A systematic review discovered that the dosage of MTF for each study was either different or not indicated at all [16].

MTF has recently been linked to mTOR signaling, a potent cellular proliferation and protein anabolism activator, via several mechanisms [14,28–30]. This results in a halting effect of tumor growth progression.

Importantly, MTF has several anti-inflammatory and antioxidant properties (Figure 3). MTF has been shown to decrease LPS-mediated inflammatory damage to the mouse nervous system [31]. MTF has also been found to protect against neuronal apoptotic cell death caused by trauma or even sepsis. This safeguard is hypothesized to be the result of decreased NF-kB translocation in the cell and leads to corresponding decreases in the production of inflammatory cytokines such as TNF-a, IL-1B, and IL-6 [32]. We observed a decrease in MCP-1, IL-1betaA, and TNF-a upon MTF treatment in GBM cells, which is in line with these hypotheses. These mediators, in turn, contribute to growth and immunosuppression.

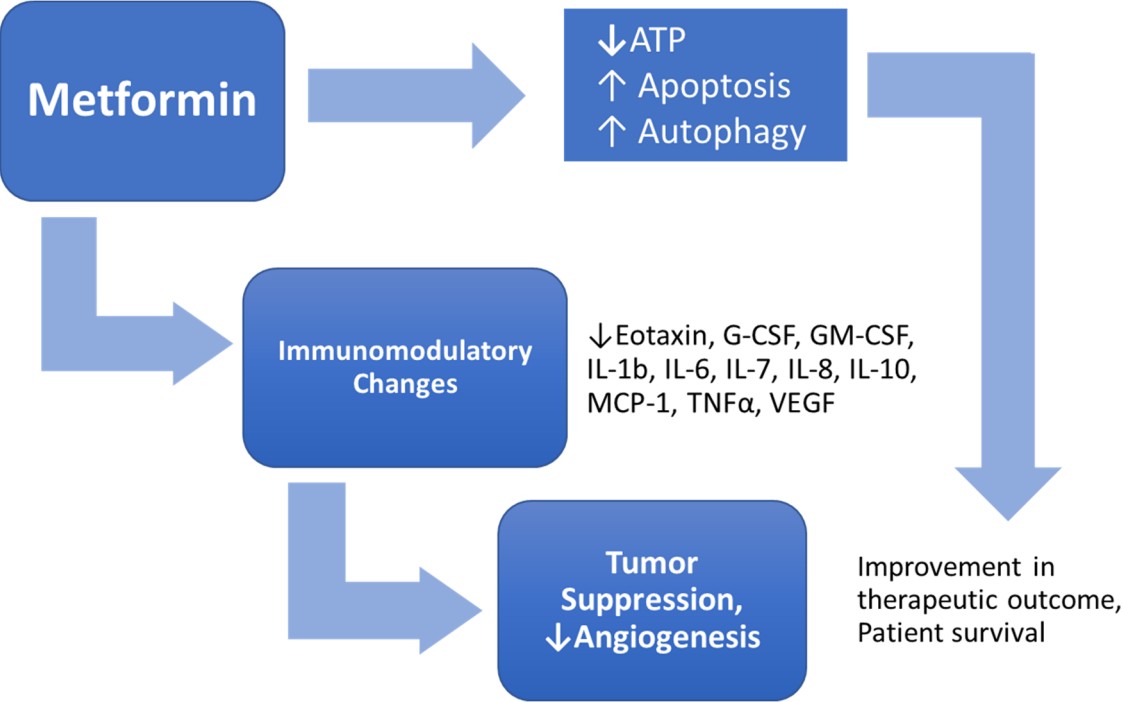

**Figure 3.** MTF has an effect on the viability of GBM cells via induction of cell death, as well as an immunomodulatory effect.

The reduction of IL-8 levels upon treatment with 50 mM MTF may be of outmost clinical importance. Endothelial cells and GBM cells present in the perivascular niche secrete enhanced levels of IL-8 and induce cancer stem-like cells to upregulate the IL-8 cognate receptors CXCR1 and CXCR2 [33], which enhance cell migration [34], as well as cell proliferation and invasion [35]. Vascular endothelial growth factor (VEGF) is the most abundant and important mediator of angiogenesis in GBM and a target of therapeutic approaches [36]. Several studies have found that MCP-1 is associated with tumor development [37], which explains its increase over time along with VEGF.

MTF can decrease the activity of M1 macrophages and allow for cell-differentiation processes to continue. This in turn causes the generation of the M2 anti-inflammatory subtype of macrophages, which can decrease reactive oxygen species (ROS) production. The inhibition of ROS production is presumed to be the result of AMPK activation by MTF [38]. ROS may cause DNA damage and mutagenesis in cells, which can be attenuated by MTF. Treatment of human fibroblasts with MTF has been shown to significantly diminish the yH2AX signal, a sensitive molecular marker of DNA damage [39]. It is unknown, however, if this reduction is simply due to lower levels of ROS, which are attenuated by MTF, or an indication of MTF's ability to sensitize cells to chemotherapy. The loss of signal indicated less DNA damage and less mutagenesis, especially when compared to paraquat cell cultures [39]. MTF can induce apoptosis in many types of cancer cells via ROS by interfering with mitochondrial physiology. Overall, the central theme regarding tumor cell arrest posits that MTF inhibits glycolysis in tumor cells, which invariably starves the cell. Different cancer cell types, nevertheless, respond in unique ways to MTF.

## 5. Conclusions

The findings from our study suggest that MTF decreases the viability of GBM cells in conjunction with an immunomodulatory effect. These effects were observed to be dose-dependent and correlated with MTF exposure time. While our study identified several mechanisms that led to the overall inhibitory effect of MTF on the human GBM cell line, further inquiry is necessary to gain a better understanding of how this in vitro study would translate into success in vivo. Furthermore, determining the optimal dosing regimen for non-labeled use of MTF would also be critical.

**Author Contributions:** Conceptualization, J.C.; methodology, D.H., R.A., J.B. and J.C.; software, D.H. and J.C.; analysis, J.C.; investigation, D.H., R.A. and J.B.; writing, D.H., R.A., K.M.R. and J.C. All authors have read and agreed to the published version of the manuscript.

**Funding:** This research received no external funding.

**Conflicts of Interest:** The authors declare no conflicts of interest.

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
