# Peer review of "Metformin Reduces Viability and Inhibits the Immunoinflammatory Profile of Human Glioblastoma Multiforme Cells"

_2571-6980, doi:10.3390/neuroglia5020006_

Round 1

Reviewer 1 Report

Comments and Suggestions for Authors

The paper : metformin reduces viability and inhibits the immunoinflammattory profile of human glioblastoma multiforme, by D Hong et al. is a study that replicates and confirms earlier invitro studies on the effect of metformin in glioblastoma cell lines (mentionned by authors in introduction (Sesen et al). The title however is misleading, it is in glioblastoma cell lines, not in glioblastoma multiforme (which is a clinical entity).

The methods and results are well presented. Infigure 3, the legend describe only graph A (VEGF) and MCP-1 (B). The graph for MCP-1 is C not B. There are no comment on graph B and D, please complete.

In discussion, which is a bit enlarged by many hypotheses, not tested in the methods and result, but still interesting, the MTF in human fibroblasts diminished yH2AX signal : please explain in this case the abbreviation yH2AX.

Metformin had been described as potential positive addition in the therapy of glioma, but other cancer as well, pancreatic, etc. However most retrospective studies show a minimal or no significnt increase in OS of glioblastoma patients, which is not surprising, most of persons taking metformin are diabetics, so if the OS is not increased, it may imply that MTX at least reduces the worse prognosis in diabetic patients, which is already great.

Therefore this article is important, even if not completely new. Metformin is a quite safe drug, compared to for example to temozolomid (probably without effect in more that 60% of patients , with unmethylated MGMT).

Author Response

Reviewer 1

The paper : metformin reduces viability and inhibits the immunoinflammattory profile of human glioblastoma multiforme, by D Hong et al. is a study that replicates and confirms earlier invitro studies on the effect of metformin in glioblastoma cell lines (mentionned by authors in introduction (Sesen et al). The title however is misleading, it is in glioblastoma cell lines, not in glioblastoma multiforme (which is a clinical entity).

-We thank the Reviewer for the kind comments, The title has been modified accordingly.

The methods and results are well presented. In figure 3, the legend describe only graph A (VEGF) and MCP-1 (B). The graph for MCP-1 is C not B. There are no comment on graph B and D, please complete.

-We apologize for this mistake. Figure 3 has been redone and the legend corrected.

In discussion, which is a bit enlarged by many hypotheses, not tested in the methods and result, but still interesting, the MTF in human fibroblasts diminished yH2AX signal : please explain in this case the abbreviation yH2AX.

-The diminished signal of phosphorylated H2AX (yH2AX) may suggest a reduction of double stranded DNA damage in human fibroblasts. It is unknown, however, if this reduction is simply due to lower levels of ROS which is attenuated by MTF or an indication of MTF ability to sensibilize cells to chemotherapy.  Studies linking this reduction to dysregulation of DNA Damage Response (DDR) pathways are needed to further understand the implications of diminished yH2AX.

The last part of the Discussion has been edited accordingly.

Metformin had been described as potential positive addition in the therapy of glioma, but other cancer as well, pancreatic, etc. However most retrospective studies show a minimal or no significnt increase in OS of glioblastoma patients, which is not surprising, most of persons taking metformin are diabetics, so if the OS is not increased, it may imply that MTX at least reduces the worse prognosis in diabetic patients, which is already great.

Therefore this article is important, even if not completely new. Metformin is a quite safe drug, compared to for example to temozolomid (probably without effect in more that 60% of patients , with unmethylated MGMT).

-Those are indeed great points, and we couldn’t agree more. We are indeed in need of more well-controlled prospective studies to clarify these effects.

Reviewer 2 Report

Comments and Suggestions for Authors

The authors attempted to explore the mechanistic details of metformin-induced inhibition of cell proliferation and immunoinflammatory response in human glioblastoma multiforme, a study of considerable timely importance. Given the timely urgency of such a study, this short communication is welcomed. However, there are several major concerns that need to be addressed to improve the manuscript and its overall contribution.

The introduction lacks clarity in articulating the rationale for the study. The final paragraphs, beginning with "We here…", requires elaboration to justify the research. Currently, it primarily reiterates known mechanisms, giving the impression of a validation study rather than an exploratory endeavor.

Unfortunately, the manuscript falls short in providing substantial novel insights into the mechanistic details as initially proposed. Particularly, the results concerning the inhibition of cell viability (Figures 1 and 2), are unsatisfactory, especially when compared to existing literature. This raises questions about the planning and assessment of the cell viability assays.

Furthermore, the minimal increase in the percentage of dead cells, even at 50nM MTF, and the lack of significant findings until 72 hours raise questions (as stated above) about the strength of the antiproliferative effects of MTF also suggested by the flow cytometry results which should be ideally showing a larger difference in the fluorescence between dead and live population. Please clarify and briefly discuss in the manuscript.

The claim of dose dependence, as stated in the abstract and elsewhere, does not align with the presented results. The rationale behind the chosen doses and timepoints should also be briefly discussed in the introduction or the discussion.

The inclusion of the ELISA approach in the study is justifiable and commendable. Please include further details on sample preparation or related citation in the methods section. Additionally, the results or findings of the fluorescence study conducted before ELISA in Section-2.2 were not presented or discussed by the authors. A brief elaboration on why this study was conducted would be beneficial.

Figures 2A-D necessitate careful revision and explanation of the observed differences. Plates A and B appear different from C and D mainly in magnification, with no clear distinctions otherwise. Improved images with proper labeling are recommended to highlight observed differences effectively.

Again, Figure 4 requires careful revision, as discrepancies are noted, such as the reduction in ATP alongside apoptosis and autophagy, contrary to expectations.

The reduction in IL-8 observed only with 50nM MTF suggests it cannot be generalized as a factor along with the rest of the immunomodulatory factors. Further discussion on why IL-8 may behave differently from other factors as observed in Fig 3 D, would be valuable. Additionally, please discuss and include citation to support the observed increase in the immunomodulatory factors under control condition after 72 hrs. as seen in Fig 3.

Please correct the typographical error "#" instead of "%" in Figure 1 where the y-axis should represent "% dead cells".

Author Response

Reviewer 2

The authors attempted to explore the mechanistic details of metformin-induced inhibition of cell proliferation and immunoinflammatory response in human glioblastoma multiforme, a study of considerable timely importance. Given the timely urgency of such a study, this short communication is welcomed. However, there are several major concerns that need to be addressed to improve the manuscript and its overall contribution.

The introduction lacks clarity in articulating the rationale for the study. The final paragraphs, beginning with "We here…", requires elaboration to justify the research. Currently, it primarily reiterates known mechanisms, giving the impression of a validation study rather than an exploratory endeavor.

-We appreciate the Reviewer’s comments. We organized the introduction so that it explains MTF effect on inflammation, then on glioblastoma, and then possible mechanisms. We end the Introduction with the aim of that paper, which for a better delivery, we have edited as follows:

The effect of MTF on GBM cell viability, showing that MTF can selectively affect human GBM tumors, and the immunomodulatory effect of MTF on GBM cells, providing evidence that inflammation is associated with GBM growth and progression [22-24], have been reported independently. We herein investigated if these two aspects of the effect of MTF on GBM were connected.

-We aimed to explore the effect on viability and tie it to an effect of MTF on inflammation, as this latter is associated with GBM growth and progress. We know from previous studies that MTF inhibits GBM cell proliferation and tumor growth within days, as well as cell death within 48 hours. From a separate study, we know that inflammation is associated with GBM growth and progression. This study builds on the current knowledge of the action of MTF in cancers and in inflammation, aiming to point if the effects of MTF through immunological mechanisms accompany the effects on GBM cell viability. Although the study is in vitro, it provides findings on specific cytokine that should point cellular pathways in which MTF may increases the overall patient survival.

Unfortunately, the manuscript falls short in providing substantial novel insights into the mechanistic details as initially proposed. Particularly, the results concerning the inhibition of cell viability (Figures 1 and 2), are unsatisfactory, especially when compared to existing literature. This raises questions about the planning and assessment of the cell viability assays.

Furthermore, the minimal increase in the percentage of dead cells, even at 50nM MTF, and the lack of significant findings until 72 hours raise questions (as stated above) about the strength of the antiproliferative effects of MTF also suggested by the flow cytometry results which should be ideally showing a larger difference in the fluorescence between dead and live population. Please clarify and briefly discuss in the manuscript.

-The Reviewer’s raises a good point. A study by Hassan et al, 2018 demonstrated a dose dependent reduction of cell viability and cell proliferation. Their experimental design consisted on treating U87 cells at Metformin doses between 1- 50 mM for 24hrs. Based on this study we chose the utilized doses. Like them we also observed a similar effect in a dose dependent manner at the 24 hour mark. Interestingly, another study [Würth et al. 2013]. observed a reduced cell viability in GBM TIC cultures in a time dependent manner. Therefore, we decided to continue to expand our time points to 72 hrs. We observed maximal cell death at 24 hours, in a dose-dependent manner at that time point. It is still unknown why this is observed, leaving us to hypothesis that this may be due to MTF ability to first target bulk cells and then annihilate the remaining cells by target GB-CSC; a point worth exploring further. Such an statement has been added to the Discussion.

The claim of dose dependence, as stated in the abstract and elsewhere, does not align with the presented results. The rationale behind the chosen doses and timepoints should also be briefly discussed in the introduction or the discussion.

-A new graph in Figure 1 has been prepared to better visualize the dose effect of MTF on GBM cell viability.

In addition, to better clarify our approach, we have edited the paragraph in Material and methods section, and it now reads

“The MTF doses and time points that were incorporated as part of our assessment align with previous study designs in which MTF treatment was associated with concentration-dependent growth inhibition [Wurth et al, 2013].”

The inclusion of the ELISA approach in the study is justifiable and commendable. Please include further details on sample preparation or related citation in the methods section. Additionally, the results or findings of the fluorescence study conducted before ELISA in Section-2.2 were not presented or discussed by the authors. A brief elaboration on why this study was conducted would be beneficial.

- The following has been added to the section 2.2. Immune response evaluation in Materials and methods.

We then proceeded to perform a multiplex ELISA using the Human Cytokine 29 Plex (Millipore) which detects simultaneously 29 cytokines (EGF, G-CSF, GM-CSF, IFN-α2, IFN-γ, IL-1α, IL-1β, IL-1ra, IL-2, IL-3, IL-4, IL-5, IL-6, IL-7, IL-8, IL-10, IL-12 (p40), IL-12 (p70), IL-13, IL-15, IL-17A, IP-10, MCP-1, MIP-1α, MIP-1β, TNF-α, TNF-β, VEGF, and Eotaxin/CCL11). Supernatants were collected from U87 cells cultures following exposure of 1mM, 20mM and 50mM MTF for 1, 24, 48, and 72 hours. Cell supernatant were collected from treated cells and centrifuged to remove cellular debris and the multiplex plate was ran on a Luminex MagPix instrument (Millipore) following manufacturer’s instructions.

The section using the NF-kB GFP plasmid has now been removed as it really did not add much to the inflammation analysis, and it was only used to determine the best timepoints and concentrations for the multiplex assay.

-We thank the Reviewer for pointing out such oversight. We have now added the following statement in the Results section regarding the LIVE/Dead assay.

Figures 2A-D necessitate careful revision and explanation of the observed differences. Plates A and B appear different from C and D mainly in magnification, with no clear distinctions otherwise. Improved images with proper labeling are recommended to highlight observed differences effectively.

-We apologize for this oversight. Plates C and D were obtained with a 40X. We have removed this figure as it is better to show the new graphs on cell viability.

Again, Figure 4 requires careful revision, as discrepancies are noted, such as the reduction in ATP alongside apoptosis and autophagy, contrary to expectations.

-We have edited Figure 4 showing the correct effects.

The reduction in IL-8 observed only with 50nM MTF suggests it cannot be generalized as a factor along with the rest of the immunomodulatory factors. Further discussion on why IL-8 may behave differently from other factors as observed in Fig 3 D, would be valuable. Additionally, please discuss and include citation to support the observed increase in the immunomodulatory factors under control condition after 72 hrs. as seen in Fig 3.

-Following Reviewer’s suggestion, the following has been added to the Discussion:

“The reduction of IL-8 levels upon treatment with 50 mM MTF may be of outmost clinical importance. Endothelial cells and GBM cells present in the perivascular niche secrete enhanced levels of IL-8 and  induce cancer stem-like cells to upregulate the IL-8 cognate receptors CXCR1 and CXCR2 [Guequen et al. 2019], which enhance cell migration [Infranger at al. 2013], as well as cell proliferation and invasion [Sharma et al. 2018]”.

“Vascular endothelial growth factor (VEGF) is the most abundant and important mediator of angiogenesis in GBM, and a target of therapeutic approaches [Weathers and de Groot 2015]”

“Several studies have found that MCP‑1 is associated with tumor development [Wang et al 2021] which explains is increase overtime along with VEGF”.

Please correct the typographical error "#" instead of "%" in Figure 1 where the y-axis should represent "% dead cells".

-We have modified Figure 1 showing two graphs, one of Number of dead cells, and a new one with % of dead cells.

Round 2

Reviewer 2 Report

Comments and Suggestions for Authors

The authors’ meticulous revision has significantly improved the manuscript, in particular, the adequate changes made in the introduction, results, and discussion.

Minor remarks/suggestions:

a. Instead of providing the correct fluorescent images, the authors chose to completely remove them. Is it possible that the differences among the groups were visually indistinct especially considering the smaller number/percent of dead cells within individual samples in a given field of view?

b. I highly recommend matching the color pattern among Figures 1A, 1B, and 1C to improve coherency. Additionally, if the authors conducted statistical analysis using the percentage data, adding error bars to Figure 2B would be ideal.

Author Response

The authors’ meticulous revision has significantly improved the manuscript, in particular, the adequate changes made in the introduction, results, and discussion.

-Such improvement is all thanks to the Reviewer’s comments and suggestions.

Minor remarks/suggestions:

  1. Instead of providing the correct fluorescent images, the authors chose to completely remove them. Is it possible that the differences among the groups were visually indistinct especially considering the smaller number/percent of dead cells within individual samples in a given field of view?

-Through the count of dead (red) in background of live (green) cells we determine the viability. This was done using ImageJ, so there is no chance of visual inaccuracy.

We can put back the image panel as a supplementary figure if the Reviewer and/or Editor thinks it is beneficial.

  1. I highly recommend matching the color pattern among Figures 1A, 1B, and 1C to improve coherency. Additionally, if the authors conducted statistical analysis using the percentage data, adding error bars to Figure 2B would be ideal.

-This is an excellent suggestion. The graphs in Figure 1 now all have the same color pattern.
